# Production and Secretion of Gelsolin by Both Human Macrophage- and Fibroblast-like Synoviocytes and GSN Modulation in the Synovial Fluid of Patients with Various Forms of Arthritis

**DOI:** 10.3390/biomedicines10030723

**Published:** 2022-03-21

**Authors:** Jessica Feldt, Martin Schicht, Jessica Welss, Kolja Gelse, Stefan Sesselmann, Michael Tsokos, Eileen Socher, Fabian Garreis, Thomas Müller, Friedrich Paulsen

**Affiliations:** 1Institute of Functional and Clinical Anatomy, Friedrich-Alexander University Erlangen-Nürnberg (FAU), Universitätsstr. 19, 91054 Erlangen, Germany; jessica.feldt@fau.de (J.F.); jessica.welss@fau.de (J.W.); eileen.socher@fau.de (E.S.); fabian.garreis@fau.de (F.G.); 2Department of Trauma Surgery and Orthopaedic Surgery, Hospital Traunstein, 83278 Traunstein, Germany; kolja.gelse@kliniken-sob.de; 3Institute for Medical Engineering, University of Applied Sciences Amberg-Weiden, 92224 Amberg, Germany; s.sesselmann@oth-aw.de; 4Institute of Legal Medicine and Forensic Sciences, Charité-Universitätsmedizin Berlin, 10117 Berlin, Germany; michael.tsokos@charite.de; 5Department of Child and Adolescent Medicine, Pediatrics I, Pediatric Rheumatology, Martin Luther University Halle-Wittenberg (MLU), 06108 Halle (Saale), Germany; thomas.gerd.mueller@bluewin.ch

**Keywords:** arthritis, gelsolin, synovial fluid, synoviocytes

## Abstract

Gelsolin (GSN) is an actin-binding protein involved in cell formation, metabolism and wound closure processes. Since this protein is known to play a role in arthritis, here we investigate how the synovial membrane with its specific synoviocytes contributes to the expression of GSN and how the amount of GSN expressed is modulated by different types of arthritis. Synovial membranes from adult healthy subjects and patients with rheumatoid arthritis (RA) and osteoarthritis (OA) are analyzed by immunofluorescence, Western blot and ELISA. Macrophage-like synoviocytes (MLS) and fibroblast-like synoviocytes (FLS) were isolated, cultured and analyzed for their potential to produce and secrete GSN. In addition, the GSN concentrations in the synovial fluid of various forms of arthritis are determined by ELISA. GSN is produced by the healthy and arthritic synovial membranes. Both forms of synoviocytes (MLS and FLS) release GSN. The results show that there is a significant reduction in GSN in the synovial fluid in adult patients with OA. This reduction is also detectable in adult patients with RA but is not as evident. In juvenile arthritis, there is a slight increase in GSN concentration in the synovial fluid. This study shows that primary MLS and FLS express GSN and that these cells, in addition to articular chondrocytes, contribute to GSN levels in synovial fluid. Furthermore, GSN concentrations are modulated in different types of arthritis. Further studies are needed to fully understand how GSN is involved in joint homeostasis.

## 1. Introduction

The synovial intima consists of macrophage-like synoviocytes (MLS), which lie superficial to the joint cavity and are capable of phagocytosis and uptake from cartilage cell metabolism products. Fibroblast-like synoviocytes (FLS) are also present, which have abundant endoplasmic reticulum, form the synovial fluid (synovia) and lie beneath the MLS [1]. MLS form in the bone marrow and migrate from embryonic progenitor cells into the synovial membrane. As recently shown, the majority of MLS form a dynamic membrane-like structure around the synovial cavity, forming an internal immunological barrier at the synovial lining and physically secluding the joint [2]. Unlike recruited monocyte-derived macrophages, which actively contribute to joint inflammation, these lining macrophages restrict the inflammatory reaction by providing a shield for intra-articular structures [2]. In addition, MLS express various surface markers such as CD11b, CD68 and CD141-3 [1,3,4]. FLS are mesenchymal-derived cells rich in rough endoplasmic reticulum and are important for collagen fibril synthesis and synovial fluid production. FLS are more numerous than MLS and secrete several characteristic factors such as surface marker CD90 (Thy-1) or vascular cell adhesion molecule 1 (VCAM-1) [1,3]. Another specific FLS marker other than urdidine diphosphoglucose dehydrogenase (UDPGD) is cadherin-11 (CDH11), which is important for cell–cell adhesion, differentiation and proliferation processes [4,5]. These synovial cells are important mediators of various joint-related diseases such as rheumatoid arthritis (RA) and osteoarthritis (OA) [6]. RA is characterized by systemic inflammation and synovitis, leading to severe joint destruction and symptoms such as pain and stiffness [7,8,9]. The contribution of the synovial membrane with its different cell types is well studied. FLS show abnormal behavior in RA such as increased survival due to anti-apoptotic gene expression. In addition, it was shown that FLS invade into the cartilage tissue of RA, leading to its degradation [4]. In OA, inflammation of the synovial tissue is also an important factor in its onset. The combination of angiogenesis and accumulation in the extracellular matrix (ECM) and synovial fibrosis with the release of TGF-β, Tissue Inhibitor of Metalloproteinase 1 (TIMP1) as well as VEGF are key points in the homeostasis of the joint [10,11,12]. The secretion of inflammatory mediators by synoviocytes directly affects chondrocytes and leads to a “fibrotic” state of the joint [10,13,14]. Both types of arthritis are common in older people. The risk of developing RA or OA increases significantly with age [9,15]. However, juvenile forms of arthritis exist. Cases in younger people as well as chronic forms of arthritis in children and adolescents are described. While RA is a unified and defined disease in adults, chronic joint inflammation consists of a group of diseases currently classified into six subtypes and a rubric for unclassifiable cases in young people [16,17]. Taking recent discussions into account, only four subtypes remain [18]. However, they do not represent a unified entity as they differ immunologically, immunogenetically, immunohistologically, as well as in terms of their pathogenesis and clinical presentation [19,20,21,22,23].

Since 1997, they have been grouped under the term juvenile idiopathic arthritis (JIA). Clinically, all forms of JIA have in common that one or more joints are inflamed for more than 6 weeks up to the age of 16 and no other known cause can be named. Only (rheumatoid factor) RF-positive polyarticular JIA is identical to RF-positive polyarthritis in adults and forms continuity. Only a small percentage of patients with joint inflammations as described above recover over time. A significant proportion of patients show changes in the structure and function of the affected joints after years of relapses and carry the disease into adulthood [24,25,26,27]. Among other things, cartilage thickness is significantly reduced in patients with JIA compared to healthy controls [28]. To date, there is no valid biomarker that can be used to determine the success of a therapy of JIA [29]. Given its potential role in the etiology and pathogenesis of JIA, particularly in relation to cartilage tissue, the actin-binding protein gelsolin currently has an unknown status in both research and pediatric rheumatology practice.

Today, many different therapeutics are known to treat arthritis successfully. These include pharmacological approaches such as non-steroidal anti-inflammatory drugs, non-pharmacological approaches such as weight reduction, dietary changes and physiotherapy, as well as intra-articular injections of therapeutic agents such as hyaluronan [30,31,32,33]. Various proteins are also being investigated for their therapeutic benefits for arthritis patients. One of these is the protein gelsolin (GSN). GSN is an actin-binding protein involved in cell mobility, cell shape, metabolism and wound healing processes [34,35]. Three isoforms are known today, with the plasmatic form (pGSN) being the most common, involved in various immune system processes as well as wound healing [34,36]. The structural data that have been established form the basis of the functional understanding of gelsolin, which have recently been summarized in more detail [34]. The molecule consists of six homologous domains, the so-called gelsolin repeat motifs (Figure 1).

In arthritis, GSN is a well-studied protein that is often decreased in blood serum and synovial fluid and therefore serves as a biomarker for these diseases [38,39]. In RA studies, GSN injection has already been shown to have a positive anti-inflammatory and chondroprotective effect [40]. Our own recently published results showed similar positive results of GSN in an OA in vitro model, as well as a significant positive effect on human primary chondrocyte survival and migration [36]. Since GSN has also been detected in synovial fluid, it is reasonable to assume that GSN is produced by synoviocytes [41]. The aim of the present study was to elucidate the site of formation and concentration of GSN in the synovial membrane and synovial fluid in order to provide a basis for possible therapeutic approaches.

## 2. Materials and Methods

### 2.1. Cell Isolation and Culture

Macrophage-like synoviocytes (MLS) and fibroblast-like synoviocytes (FLS) were isolated from human synovial membrane samples. These were collected from patients with osteoarthritis undergoing knee replacement surgery at the University Hospital Erlangen-Nuremberg, with institutional review board approval (No.: 3555) in compliance with the Declaration of Helsinki (♀ = 2, 67 and 73 years; ♂ = 1, 82 years old). For FLS isolation out of osteoarthritic synovial membranes, tissue was isolated from the whole fat-containing environment and minced. Tissue was then incubated in 1.5 mg/mL Dispase (Roche) in Dulbecco’s Modified Eagle’s Medium (DMEM) containing 10% FCS and 1% penicillin/streptavidin (Pen/Strep) for 2 h at 37 °C on a shaker. The cell suspension was filtered through a 100 µm cell strainer and centrifuged at 1200 rpm for 10 min. The cells were resuspended and seeded into a small cell culture flask. For MLS isolation, the tissue was cut into pieces and incubated with trypsin on a shaker at 37 °C for 20 min. The reaction was stopped with FCS-containing medium and the tissue was washed once with PBS. The tissue was then incubated for 4 h at 37 °C with 1.1 mg/mL collagenase. The cell suspension was filtered with a 70 µm cell strainer and centrifuged at 2000 rpm for 5 min. The collagenase solution was removed and the cells were seeded in a small cell culture flask in DMEM containing 10% FCS and 1% Pen/Strep.

### 2.2. Synovial Fluid Samples

Healthy human synovial fluid (SF) (*n* = 13) was obtained from donors, 4 female and 9 male, aged 29–80 years from the Department of Forensic Medicine, Halle, Germany, from autopsy cases with unnatural death not affecting the joints. The OA-SF and RA-SF (each *n* = 20) were from two different sources: 10 OA samples (7 females and 3 males, aged 64–87 years) and 10 RA samples (9 females and 1 male, aged 39–76 years) were obtained from patients undergoing total knee arthroplasty at the Department of Trauma Surgery, Erlangen University Hospital. Adolescent arthritis—SF (*n* = 18 each) were obtained from patients (8 female and 10 male, aged 3–15 years) in the Department of Paediatrics and Adolescent Medicine, Paediatrics I, Paediatric Rheumatology, Children’s Hospital, Halle (Saale), Germany (Table 1). The diagnosis of OA was based on clinical and radiographic examinations according to standard criteria, and the diagnosis of RA was made in patients who met the American College of Rheumatology/European League Against Rheumatism classification criteria. Subtypes of juvenile arthritis (JIA) from children and adolescent patients (Table 1) comprised oligoarticular OA (JIA-OA; *n* = 7), extended oligoarthritis (eOA; *n* = 2), enthesitis-associated (EAA, *n* = 1), psoriatic arthritis (PsA; *n* = 2), acute yet unclassified monoarthritis (AMA; *n* = 3), and polyarticular, rheuma factor negative arthritis (PolyRF; *n* = 2). Each patient/responsible gave informed consent prior to the procedure and the institutional ethics committee approved this study (Ref.No. 3555; FAU Erlangen-Nuremberg, Ref.No. 2013–30; MLU Halle-Wittenberg). The SF samples were transferred directly into a 1.5 mL Eppendorf tube and immediately frozen on dry ice. All samples were then transported on dry ice from the clinic to the laboratory and stored at −80 °C until analysis. Samples were centrifuged at 13,000 rpm for 5 min before use. The supernatants were frozen or subsequently analyzed by enzyme-linked immunosorbent assay (ELISA).

### 2.3. Immunohistochemically Staining

The synovial membrane samples from body donors of the Friedrich-Alexander University Erlangen-Nuremberg which are embedded in paraffin were deparaffinized by incubating the slides in xylene for 10 min. Subsequently, the samples were incubated in a second xylene solution for 20 min before continuing the deparaffinization with ethanol (100%/100%/96%/80%/70%). After washing the slides twice with distilled water, they were incubated in 3% H2O2 for 10 min at room temperature. The slides were then washed three times with distilled water before being heated in citrate buffer (pH 6) for 10 min and then cooled to room temperature for at least 1 h. After a 10 min incubation with trypsin, the slides were washed with TBS-T and treated with normal goat serum for 20 min at room temperature. An avidin/biotin blocking kit (BioLegend, SIG-31126) was used for 10 min each before the primary antibody (GSN, Santa Cruz, 48749, 1:50) was added overnight at 4 °C. The next day, the slides were washed in TBS-T before the secondary antibody (DAKO, 86048, 1:200) was added for 1 h at room temperature. After incubation, the slides were washed three times and treated directly with an ABC kit (Vector Laboratories, PK-6100) for 1 h at room temperature before staining with an AEC solution (DAKO, K3461). The staining was stopped with distilled water and counterstained with hemalum before the slides were covered.

### 2.4. Immunofluorescence Staining

Cells were cultured as previously described, but on a coverslip pre-coated with FCS. After the cells reached the required confluence, they were washed three times with TBS-T and blocked with Blotto (2 g milk powder in 100 mL PBS with 100 µL Tween 20) for 1 h at room temperature. After washing the cells, the primary antibody (GSN: Santa Cruz 48749; MLS = CD68: DAKO M0876; FLS = CDH11: R&D Systems MAB1790) was added at 4 °C overnight. Cells were washed three times with TBS-T and secondary antibody (goat α-rabbit; Invitrogen 1812158; goat α-mouse; Invitrogen 189294) was added for 1 h at room temperature. After washing the cells again, DAPI was added for 10 min before the cells were covered for analysis.

### 2.5. Western Blot

FLS and MLS samples were used at a concentration of 25 µg, mixed with 5 µL RSB (resuspension buffer), incubated (5 min, 100 °C) and separated in a 15% SDS gel. The gel was blotted onto a nitrocellulose membrane (GE Healthcare, #10600008). The membrane was blocked in 5% BSA-PBS-T for 1 h at RT before the primary GSN antibody (Sigma Aldrich; GS-2C4, #G4896, 1:1000) was added overnight (4 °C). After washing in PBS-T, the secondary antibody (Cell Signalling; #7076S) was added for 2 h at RT. The membrane was washed before using an ECL mixture (Millipore; #WBKIS0500) for chemiluminescence detection. For GAPDH, the membrane was treated with stripping buffer (65 °C, 45 min), washed in TBS-T and blocked in 5% milk powder TBS-T before the GAPDH antibody (Santa Cruz, #Sc-365062, 1:2000) was added overnight (4 °C). After washing, the secondary antibody (Cell Signalling; #70769, 1:5000) was added (2 h, RT). An ECL mixture was used for chemiluminescence detection.

### 2.6. ELISA

For the analysis of GSN content in FLS and MLS (♀ = 2, 67–73 years; ♂ = 1, 82 years old) and synovial fluid samples (10 OA samples (7 females and 3 males, aged 64–87 years) and 10 RA samples (9 females and 1 male, aged 39–76 years)) of healthy controls and arthritis patients, the Cloud-Clone (SEA372 Hu) ELISA was used according to the given protocol. For this, protein was isolated from the synoviocytes using 300 µL of Triton buffer containing 0.2% protease and 0.2% phosphatase inhibitors. The cells were incubated on ice for 30 min. After centrifugation (13,000 rpm, 4 °C, 5 min), the supernatant was decanted and protein concentration was measured by Bradford assay. The same amount of protein was used for the comparison of FLS and MLS as well as for the synovial fluid samples.

### 2.7. Statistics

The results were analyzed using t-tests and chi-square tests. Results are expressed as the mean ± SEM and *p*-values below 0.05 are considered significant. Calculations and visualizations were performed using GraphPad Prism 6 (GraphPad Prism Software).

## 3. Results

### 3.1. MLS and FLS React Positively with an Antibody against Gelsolin

Sections through the healthy synovial membrane show a strong positive reactivity in the MLS of the synovial membrane and also a weak positive reaction in the FLS. Single cells in the subsynovial connective tissue also show a positive reaction (Figure 2).

### 3.2. Both Isolated and Cultured MLS and FLS Produce Gelsolin

To prove that the GSN detected was indeed produced by synoviocytes, we isolated MLS and FLS from the human synovial membrane of OA patients. The cultured cells were characterized with typical markers and stained by immunofluorescence to detect GSN (Figure 3). FLS subjectively show lower GSN fluorescence intensity than MLS.

To verify this aspect, we analyzed the GSN concentration of cultured MLS and FLS separately by ELISA (Figure 4). Interestingly, both MLS and FLS isolated from OA synovial membranes showed similar GSN concentrations with an average of 0.09 µg/mL (MLS: 0.0895 µg/mL; FLS: 0.0905 µg/mL).

### 3.3. Gelsolin Is Also Detectable in the Synovial Membrane of Patients with RA and OA

Gelsolin can also be localized in the synovial membrane of patients with rheumatoid arthritis (RA) and osteoarthritis (OA) (Figure 5). A heterogeneous distribution of GSN can be seen in the synovial membrane of patients with RA (Figure 5 upper row of images). The antibody against gelsolin reacts positively, especially in the lower cell layers of the synovial membrane as well as in the subsynovial connective tissue. CD69/CDH11 double reaction to localize MLS and FLS is weak but positive (Figure 5 upper row of images). The GSN antibody reaction in synovial membrane sections of OA patients do not differ from those of body donors shown in Figure 2 with regard to GSN distribution (Figure 5 lower row of images).

### 3.4. Gelsolin Is Secreted into Synovia and Is Significantly Reduced in Cases of OA

Quantitative analysis of synovial fluid samples from healthy individuals and patients with RA and OA shows that GSN is detectable in all samples examined (Figure 6).

In RA patients, there is no significant reduction in GSN compared to healthy controls. In patients with OA, the GSN concentration in the synovial fluid is also significantly reduced (Figure 6B). The synovial fluid of healthy individuals has an average GSN concentration of 7.89 ± 1.15 ng/mL (Figure 6; Table 2. The GSN concentration in RA patients is slightly lower on average at 7.11 ± 0.93 ng/mL (Figure 6B; Table 1). Analysis of the synovial fluid of OA patients reveals a significant reduction to 4.49 ± 0.22 ng/mL compared to healthy subjects (*p*-Value: 0.0059) and RA patients (*p*-Value: 0.0037) (Figure 6B; Table 2).

### 3.5. Patients with Different Subtypes of Juvenile Arthritis Have Higher GSN Concentrations in the Synovial Fluid Than Older Patients or Healthy Adults

A total of 17 patients with different forms of JIA were included in the ELISA measurements of synovial fluid (Table 1). It can be seen that the average GSN value in JIA-OA (10.26 ng/mL) is higher than in other JIA subtypes with the exception of one child with JIA from type EAA (Table 3).

Compared with the GSN concentration in the synovial fluid of healthy adults (7.89 ng/mL) and adult patients with RA (7.11 ng/mL) or OA (4.49 ng/mL), this value is slightly higher at 10.26 ng/mL and more than twice as high compared with OA in adults (Table 2 and Table 3).

The analysis showed that the average value (10.26 ng/mL) is higher than in other subtypes of JIA (Table 3). Therefore, we were interested in the distribution of low GSN values among the JIA subtypes (Table 4).

There appears to be a correlation between the subtype of JIA (persistent oligoarthritis versus all other forms of JIA) and the level of GSN measured. In our series of measurements with a clear assignment to a certain subtype of JIA, patients with persistent oligoarthritis had significantly higher GSN values (≥10 ng/mL) and all other subtypes of JIA had significantly lower values (*p* = 0.05). (Table 4).

## 4. Discussion

Our results show that GSN is produced by both MLS and FLS of the synovial membrane and secreted into the synovial fluid. We found a significant reduction in GSN in the synovial fluid of OA patients (Figure 6B). GSN has gained increasing attention in recent years as this protein has been implicated in various diseases [34]. In addition to joints, GSN has been detected in lung, heart, skeletal muscle and a number of other tissues [42]. For chondrocytes of articular cartilage, GSN has been shown to have a protective effect in simulating regenerative processes and differentiation in cartilage [36,43]. Therefore, the presence of GSN in synovial fluid is important for maintaining the health of cartilage and joint. Our results (Figure 6) confirm previous studies that GSN is present in synovial fluid and decreases in patients with rheumatoid arthritis (RA) [44]. However, in this study, the reduction in GSN was not significant as we also found single RA patients with increased GSN concentrations (Table 3). Whether this is a particular stage of RA or a particular form of RA remains to be investigated. In contrast to previous studies in which OA patients had higher GSN concentrations in the synovial fluid than RA patients, ELISA analysis in the previous study reveals significantly lower GSN concentrations in the OA group compared to RA patients and healthy subjects (Figure 6B) [45]. In contrast, Western blot analysis (Figure 6A) also shows a slight decrease in GSN concentration in the synovial fluid of OA patients compared with the synovial fluid of RA patients or healthy subjects. These differences may be caused by different stages of OA, the age of the patients/healthy subjects studied and the method used. Osborn et al. [39] showed that the plasma isoform of gelsolin is decreased in the plasma of patients with rheumatoid arthritis compared with healthy controls. We were able to confirm this result only in rudimentary form and consider it possible that the concentrations of gelsolin in the synovial fluid depend on how far the disease has progressed and whether there has been local consumption of this potentially anti-inflammatory protein in the inflamed joint. The importance of actin organization in controlling chondrocyte phenotype is well known. In hypertrophic chondrocytes, an actin-binding gelsolin-like protein called adserverin has already been identified. Overexpression of adseverin in non-hypertrophic chondrocytes leads to a restructuring of the actin cytoskeleton, a change in cell morphology [42]. These changes are mediated by the ERK1/2 and p38 kinase pathways. It is conceivably that gelsolin functions similarly. In addition to gelsolin itself, other members of the gelsolin family, such as Flightless I, are also involved in the hypertrophy and catabolism of chondrocytes and thus directly in arthritis itself [43]. Given the many members of the gelsolin family involved and their seemingly opposing roles in arthritis, further research on this topic needs to be conducted to understand the involvement of gelsolin in arthritis. It is conceivable that gelsolin functions in a similar manner. In addition to gelsolin itself, other members of the gelsolin family, such as Flightless I, are also involved in the hyperthrophy and degradation of chondrocytes and thus directly in arthritis itself [46]. Given the many members of the gelsolin family involved, such as scinderin (adseverin) [47] as well, and their seemingly opposing roles in arthritis, further research on this topic is needed to understand the involvement of gelsolin in arthritis.

A surprising finding is that the GSN concentration in patients with juvenile arthritis is higher compared to adults with RA or OA and compared to synovial fluid from healthy subjects. As the number of samples analyzed in this study is low, this finding has to be further approved. Nevertheless, a possible explanation could be that (1) children and adolescents generally have a higher GSN concentration in the synovial fluid. Since no synovial fluid samples of an age-matched group of people were available, this hypothesis could not be tested. (2) Artificial admixture of blood may have occurred during joint puncture, which could have increased the GSN concentration in the synovial fluid. (3) The stage and duration of JIA could have an influence on the higher GSN concentration and (4) the GSN concentration could depend on the subtype of JIA.

This last hypothesis in particular arises from the finding that the measured GSN concentrations are not evenly distributed in the subgroups of JIA. Apparently, there is a trend towards higher GSN values in persistent oligoarthritis and lower values (below 10 ng/mL) in the other subtypes. The only exception is the value of 10.7 ng/mL in EAA. In this context, it is important to note that there can be overlap between the two JIA subtypes, i.e., enthesitis can also occur in oligoarthritis and is not always a sign of juvenile spondyloarthritis. Nevertheless, an elevated gelsolin level in JIA persistent oligoarthritis seems somewhat plausible: assuming that GSN exerts a protective effect on the joint in case of inflammation, higher levels should be found in the JIA subtype with the best prognosis. The good prognosis of persistent oligoarthritis of JIA for joint function is well established. Larger samples of juvenile arthritis considering the JIA subtype and samples from children without inflammatory joint disease could provide further information.

We know from previous studies that GSN is expressed in chondrocytes of articular cartilage. However, the concentration of GSN produced by chondrocytes in the synovial fluid cannot be explained [36]. It was therefore reasonable to assume that synoviocytes, as the inner layer of the synovial membrane, are involved in the production and secretion of GSN into the synovial fluid. With the present studies, we can show that both MLS and FLS are involved in GSN production (Figure 3 and Figure 4). Due to the cell-shaping and actin-binding function of GSN, it makes sense that this protein is produced in the synovial membrane, which has high concentrations of actin. It is hypothesised that cytoskeletal restructuring in the synovial membrane may contribute to the development of arthritis [48] and may be a consequence of reduced GSN levels in synovial fluids from arthritis patients. As GSN is highly involved in regulating the cytoskeleton by interacting with actin and shows various activation statuses depending on its surrounding environment [49], additional experiments must focus on the effect of synovial fluid on GSN activity. Both MLS and FLS contribute to inflammation processes in the arthritic joint due to changes in cell metabolism [50]. GSN targets cell metabolism by involvement in signal transduction and transcriptional processes and is also involved in inflammatory cascades [51]. Therefore, it is suspected that reduced levels of GSN are involved in the development of inflammatory processes in the synovial membrane, especially in the synoviocytes. Further investigation is required to focus on how GSN affects joint homeostasis and the significance of reduced GSN expression in adult arthritis and whether the increased GSN levels in JIA should be understood as a response to the onset of arthritis.

Limitations of the present study include the small sample size of juvenile patients. Although samples were carefully selected from all patients who had no history of disease, there is a possibility that other factors (diseases and medications) may influence the results.

In summary, we present here evidence of GSN expression and secretion by both forms of synoviocytes (MLS and FLS) into synovial fluid and show that there is a significant reduction in GSN in synovial fluid in adult patients with OA. This reduction is detectable in adult patients with RA, but not as clearly visible. In juvenile arthritis, there is a slight increase in GSN concentration in synovial fluid. The subtypes seem to differ with respect to the gelsolin, which shows that it is not a uniform entity. This has to be confirmed in further experiments. Assuming that gelsolin can serve as a new prognostic marker for treatment (e.g., therapy more or less aggressive), further studies are needed to investigate the ability of synoviocytes to express GSN and to determine the role of GSN in arthritis.

## Figures and Tables

**Figure 1 biomedicines-10-00723-f001:**
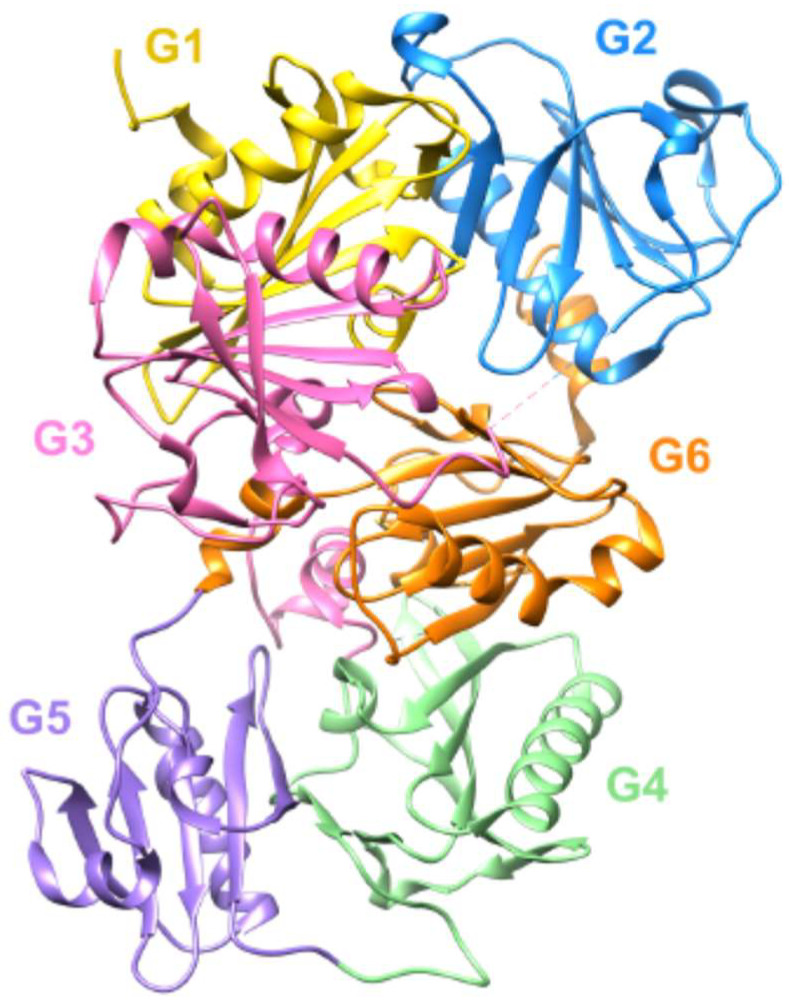
Structure of the six homologue domains of GSN. Schematic representation of the structure of Ca-free human gelsolin (PDB ID code: 3FFN) [37].

**Figure 2 biomedicines-10-00723-f002:**
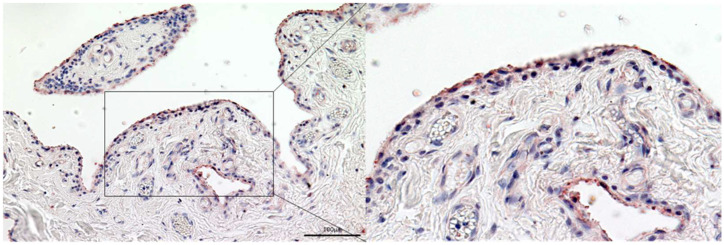
MLS and FLS react positively with an antibody against gelsolin. Section through the healthy synovial membrane. The antibody against gelsolin reacts intensively positive (red) with the MLS of the synovial membrane and also shows a weak positive reaction in the FLS and with individual cells in the subsynovial connective tissue. Counterstain: hemalum. Inset magnification ×10.

**Figure 3 biomedicines-10-00723-f003:**
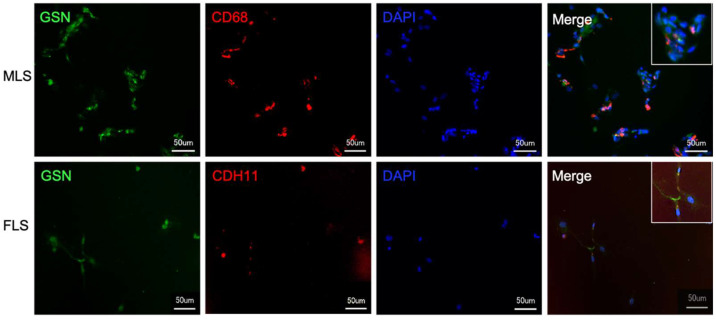
Isolated MLS and FLS producing gelsolin. Isolated and cultured macrophage-like synoviocytes (MLS) and fibroblast-like synoviocytes (FLS) show a positive antibody response for cell-specific markers (MLS = CD68 (red); FLS = CDH11 (red)) and for gelsolin (GSN—green) by immunofluorescence.

**Figure 4 biomedicines-10-00723-f004:**
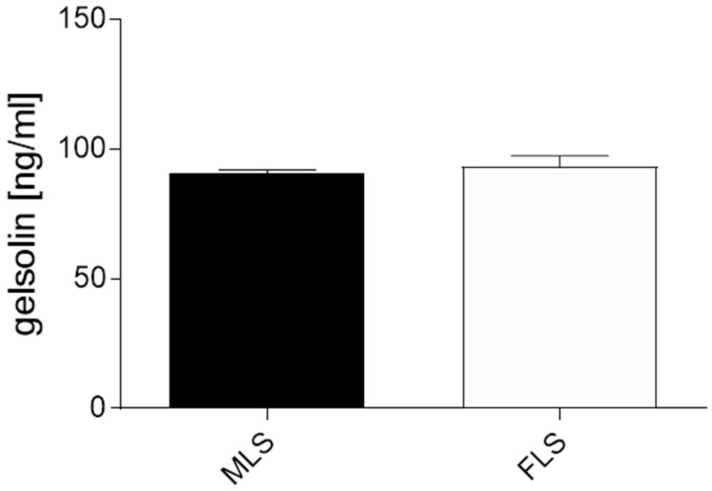
ELISA analysis of MLS and FLS. Isolated and cultured MLS and FLS isolated out of the synovial membrane of osteoarthritis patients produce GSN at approximately the same concentration (0.09 µg/mL). ELISA analysis of gelsolin in MLS and FLS, *t*-test.

**Figure 5 biomedicines-10-00723-f005:**
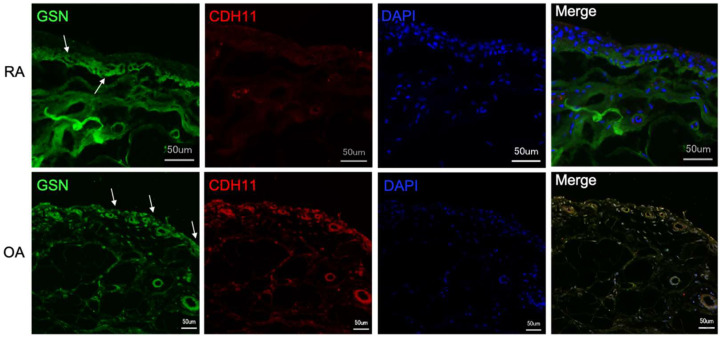
Gelsolin detection in synovial membranes and fluids of RA and OA patients. Immunofluorescence detection of gelsolin (GSN—green, white arrows) in the synovial membrane of patients with rheumatoid arthritis (RA) and osteoarthritis (OA). Paraffin-embedded tissue sections are stained with antibodies against CD68 (red) and CDH11 (red) to localize macrophage-like synoviocytes (MLS) and fibroblast-like synoviocytes (FLS) as well as gelsolin (GSN) to localize GSN. GSN is detectable in the subepithelial tissue of the RA synovial membrane (top row) and in the superficial cell layers of the synovial membrane of OA patients (bottom row).

**Figure 6 biomedicines-10-00723-f006:**
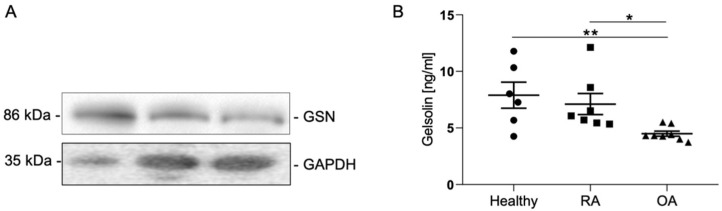
Detection and quantification of gelsolin in synovial fluid from healthy subjects and RA and OA patients. (**A**) Western blot analysis of synovial fluid (SF) from a healthy control subject and patients with RA and OA. (**B**) Gelsolin ELISA of SF samples from healthy subjects, RA and OA patients, *t*-test, * *p* < 0.5; ** *p* < 0.01.

**Table 1 biomedicines-10-00723-t001:** Explanations of JIA subtypes and number of patients (8 female, 9 male, aged 3–15).

Abbreviation	Subtyp Juvenile Arthritis (JIA)	Number of Patients
JIA-OA	Oligoarticular (oligoarthritis)	7
eOA	Extended oligoarthritis	2
EAA	Enthesitis associated	1
PsA	Psoriatic arthritis	2
AMA	Acute, yet unclassified monarthritis	3
PolyRF-	polyarticular, RF negative	2

**Table 2 biomedicines-10-00723-t002:** Gelsolin concentration in synovial fluids.

	Healthy (*n* = 6)	RA (*n* = 7)	OA (*n* = 8)
Ø ng/mL ± SEM	7.89 ± 1.15	7.11 ± 0.93	4.49 ± 0.22
Individual ng/mL	7.57	5.35	4.32
	11.79	8.58	4.32
	8.02	6.52	5.41
	7.23	12.12	3.73
	4.26	5.70	4.01
	5.68	5.44	4.41
		6.07	5.50
			4.27

**Table 3 biomedicines-10-00723-t003:** GSN values in different subtypes of JIA.

JIA Subtype	OA	eOA	EAA	PsA	AMA	PolyRF-
Ø ng/ml	10.3	7.9	10.7	7.1	8.1	6.7
Individual ng/mL	10.7	9.1	10.7	8.2	8.4	6.8
	9.3	6.8		6.0	7.1	6.6
	12.9				8.8	
	5.6					
	10.0					
	13.3					

**Table 4 biomedicines-10-00723-t004:** Distribution of GSN values among patients with/without JIA oligoarthritis (*n* = 17).

Patients	GSN < 10 ng/mL	GSN ≥ 10 ng/mL	Total	Frequency
With Oligoarthritis	2	4	6	28.6%
Without Oligoarthritis	10	1	11	90.9%
Total	12	5	17	70.6%

## Data Availability

Please contact authors for data requests (M.S.; email: martin.schicht@fau.de).

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
