# Peer review of "Production and Secretion of Gelsolin by Both Human Macrophage- and Fibroblast-like Synoviocytes and GSN Modulation in the Synovial Fluid of Patients with Various Forms of Arthritis"

_biomedicines, 2022, doi:10.3390/biomedicines10030723_

Round 1

Reviewer 1 Report

This is an interesting article, however number of issues need to be addressed before this can be published:

Major weakness is that authors need to better highlight the novelty and significance of their data. At the moment numbers of patient samples used are low and its hard to draw meaningful conclusions. They should also have limitations sections at the end of discussion and future direction for this work. 

Minor issues:

Page 2 Introduction - break up the text into 2 paragraphs. 

Please don't start sentences with "It"

Can authors speculate which domain of Gelsolin is important for its function in arthritis?

Authors mention briefly plasma gelsolin and its role in actin remodelling however they do not discuss this in terms of bigger picture. For example, they find that Gelsolin is reduced in arthritis but dont explain this in terms of its actin scavenging role where Gelsolin has been shown to be secreted during tissue injury and inflammation to sequester actin to prevent toxicity. They also don't mention how their work fits with other members of the gelsolin family, especially Flightless protein - which is also secreted, and increased in response to tissue injury and inflammation and scarring (in burns, in psoriasis, in epidermolysis bullosa) and was suggested to compete with Gelsolin for actin scavenging system function. 

How do the levels of secreted Gelsolin detected in synovial fluid compare to those reported in literature for plasma Gelsolin post burn injury for example. Authors should discuss their results in terms of acute vs chronic conditions which may explain differences they are seeing in juvenile vs adult patients with chronic conditions.

Figure 6, authors should include a loading control band for their WB. 

Top of page 9, instead of saying there appears to be a correlation between....can authors calculate the correlation coefficient?

Can authors describe in the discussion which pathway or function of secreted Gelsolin might be actin via in the arthritis?  

Author Response

Point-by-point response to the reviewers' comments

Reviewer #1:

  1. At the moment numbers of patient samples used are low and its hard to draw meaningful conclusions. They should also have limitations sections at the end of discussion and future direction for this work. 

Answer: The limitations sections has been changed accordingly, line 414.

Limitations of the present study include the small sample size of juvenile patients. Although samples were carefully selected from all patients who had no history of disease, there is a possibility that other factors (diseases, medications) may influence the results.

  1. Introduction - break up the text into 2 paragraphs

Answer: The sentence has been changed accordingly.

  1. Please don't start sentences with "It"

Answer: The sentence has been changed accordingly, line 338

  1. 4. Can authors speculate which domain of Gelsolin is important for its function in arthritis?

Answer: The sentence has been changed accordingly, line 359

Osborn et al. (2008) showed that the plasma isoform of gelsolin is decreased in the plasma of patients with rheumatoid arthritis compared with healthy controls. We were able to confirm this result only in rudimentary form and consider it possible that the concentrations of gelsolin in the synovial fluid depend on how far the disease has progressed and whether there has been local consumption of this potentially anti-inflammatory protein in the inflamed joint.

  1. Figure 6, authors should include a loading control band for their WB

Answer: We apologize that the control (GAPDH) was not shown, and we have changed the figure.

  1. Top of page 9, instead of saying there appears to be a correlation between. can authors calculate the correlation coefficient?

Answer: We thank the reviewer for this comment. The limitation is the small sample size of juvenile patients Our results are partly a first description. It is clear that the sample size is not sufficient to ensure a sound statement such as correlation coefficients. But we have listed sample size and significance (P-value) to support the statement.

  1. Can authors describe in the discussion which pathway or function of secreted Gelsolin might be actin via in the arthritis?

Answer: The importance of actin organization in controlling chondrocyte phenotype is well known. In hypertrophic chondrocytes, an actin-binding gelsolin-like protein called adserverin has already been identified.  Overexpression of adseverin in nonhypertrophic chondrocytes leads to a restructuring of the actin cytoskeleton, a change in cell morphology (Nurminsky et al., 2006). These changes are mediated by ERK1/2 and p38 kinase pathways. It is conceivably the Gelsolin functions similarly.

The sentence has been changed accordingly, line 364.

Reviewer 2 Report

This is an interesting manuscript. However, the manuscript is not suitable for publication in the present form. Some concerns are enlisted below

  1. Sample size is too small with high heterogeneity. I suggest focusing on the RA and OA group, then compare the joint fluid/serum gelsolin concentration among the OA/RA patients with different severity. Sample size calculation should be performed.
  2. Does the detection of gelsolin concentration harbor any diagnostic or prognostic value for the disease? The authors should try to demonstrate the pertinent values. 

Author Response

Point-by-point response to the reviewers' comments

Reviewer #2:

  1. Sample size is too small with high heterogeneity. I suggest focusing on the RA and OA group, then compare the joint fluid/serum gelsolin concentration among the OA/RA patients with different severity. Sample size calculation should be performed.

Answer: It is clear that the sample size is not sufficient to guarantee a sound conclusion, but it should provide a first insight to drive further research. Our research group has many years of experience in the study of articular cartilage (Rösler et al. 2010 Arthritis Rheum) and other tissues. The samples used here were collected over a very long period of time (more than 3 years). It should not be underestimated how difficult it is to obtain such sample material in Germany, which is extremely complex from both an ethical and medical point of view. The collection of synovial fluid from "healthy" and adolescent patients has not yet been replicated by any other research group. It took us a long time to establish this methodologically with our clinical cooperation partners. In contrast, the collection of synovial tissue and fluid from patients with OA and RA is not as time consuming.

The fact that we only examined one patient each with OA and RA by WB-PCR is due to our previous experience, because ELISA is the gold standard for protein quantification. The number of samples was sufficient for our conclusion, as we did not expect a different conclusion even with a higher number of samples.

  1. Does the detection of gelsolin concentration harbor any diagnostic or prognostic value for the disease? The authors should try to demonstrate the pertinent values.

Answer: We thank the reviewer for this comment. Our results are partly a first description. It is clear that the sample numbers are not sufficient to guarantee a sound statement. It is intended to provide an initial insight to drive further research.

Reviewer 3 Report

The manuscript “Production and secretion of gelsolin by both human macro- phage- and fibroblast-like synoviocytes and GSN modulation in the synovial fluid of patients suffering from various forms of arthritis “describes that both MLS and FLS synoviocytes express gelsolin and contribute to gelsolin levels in synovial fluid. There is a reduction of gelsolin in synovial fluid in OA and RA patients, but increase in JA patients.

  1. This is an interesting paper to discover the GSN secretion from MLS and FLS and different concentration of GSN in different arthritis. The explanation is not quit clear. The authors find GSN reduction in synovial fluid of patients with OA, but not much reduction in patients with RA and the authors think this is due to inflammatory process. But in general RA has more inflammation then OA.
  2. Osborn et al (Ref. 38) describe that Decreased levels of the gelsolin plasma isoform in patients with rheumatoid arthritis. The authors should discuss the discrepancy in synovial fluid and plasma to clarify the phenomenon.

Minor:

  1. 2 is not clearly shown the staining wit anti-GSN antibody, use arrow for the clear explanation
  2. 4 shows the ELISA analysis of MLS and FLS. The authors should use other cells for the control.
  3. The authors should increase the numbers of OA, RA, and JA to provide more clear results.

Author Response

Point-by-point response to the reviewers' comments

Reviewer 3: 

  1. This is an interesting paper to discover the GSN secretion from MLS and FLS and different concentration of GSN in different arthritis. The explanation is not quit clear. The authors find GSN reduction in synovial fluid of patients with OA, but not much reduction in patients with RA and the authors think this is due to inflammatory process. But in general RA has more inflammation then OA.

Answer: We thank the reviewer for this comment. See point 4 of the answer to Reviewer #1. Osborn et al. (2008) showed that the plasma isoform of gelsolin is decreased in the plasma of patients with rheumatoid arthritis compared with healthy controls. We were able to confirm this result only in rudimentary form and consider it possible that the concentrations of gelsolin in the synovial fluid depend on how far the disease has progressed and whether there has been local consumption of this potentially anti-inflammatory protein in the inflamed joint. The sentence has been changed accordingly, line 370

  1. Osborn et al (Ref. 38) describe that Decreased levels of the gelsolin plasma isoform in patients with rheumatoid arthritis. The authors should discuss the discrepancy in synovial fluid and plasma to clarify the phenomenon.

Answer: We thank the reviewer for this comment. We hope that the improved paragraph (see comment 1) should remove any ambiguities.

  1. Is not clearly shown the staining wit anti-GSN antibody, use arrow for the clear explanation

Answer: We thank the reviewer for this tip. We have modified the figure.

  1. Figure 4 shows the ELISA analysis of MLS and FLS. The authors should use other cells for the control.

Answer: We thank reviewer #3 for this hint but cannot follow here. We have just taken these cells, because the aim was to detect gelsolin in the two synovial membrane cell types. With ELISA, there is no need to carry controls because it is a quantitative analysis. We used the same ELISA to quantify ocular surface tissues and tears from dry eye patients in 2018 (Wittmann et al. 2018 Sci Rep – Plasma gelsolin promotes re-epithelialization.

  1. The authors should increase the numbers of OA, RA, and JA to provide more clear results.

Answer: We absolutely agree to the reviewers comment. We would like to do that, but as reviewer #2 explained, we have collected more than 3 years for the samples we used. We have explained the reasons for this in detail for reviewer #2 and ask reviewer #3 to look at this section. Please believe that we would have liked to use more samples if this had been possible.

Round 2

Reviewer 1 Report

The authors have addressed all main concerns adequately and this should be now accepted for publication. They clearly include the limitations of the study and their conclusions are appropriate for the data obtained. 

Out of interest, it would be nice to include a sentence or two in the discussion on whether other members of the Gelsolin family are involved in the actin scavenging in the arthritis - eg Flightless which is also a secreted member of this family. And if authors believe there could be a compensatory mechanism between the different members of the family. Well Done! 

Author Response

Point-by-point response to the reviewers' comments

Reviewer #1:

  1. Out of interest, it would be nice to include a sentence or two in the discussion on whether other members of the Gelsolin family are involved in the actin scavenging in the arthritis - eg Flightless which is also a secreted member of this family. And if authors believe there could be a compensatory mechanism between the different members of the family.

Answer: Dear reviewer, we thank you for the reference and we have incorporated this point accordingly in the discussion. Line 369

Reviewer 2 Report

Although not perfect, the authors provide a reasonable defense for my concerns. 

Author Response

(The authors gave the same response as above.)

Reviewer 3 Report

ok for revision

Author Response

(The authors gave the same response as above.)
